# Variable Energy Fluxes and Exact Relations in Magnetohydrodynamics Turbulence

Mahendra Verma [1,*,†] , Manohar Sharma [1,†] , Soumyadeep Chatterjee [1,†] and Shadab Alam [2,†]

1   Department of Physics, Indian Institute of Technology, Kanpur 208016, Uttar Pradesh, India; kmanohar@iitk.ac.in (M.S.); soumyade@iitk.ac.in (S.C.)
2   Department of Mechanical Engineering, Indian Institute of Technology, Kanpur 208016, Uttar Pradesh, India; shadab@iitk.ac.in
*   Correspondence: mkv@iitk.ac.in; Tel.: +91-5122598515
†   These authors contributed equally to this work.

**Abstract:** In magnetohydrodynamics (MHD), there is a transfer of energy from the velocity field to the magnetic field in the inertial range itself. As a result, the inertial-range energy fluxes of velocity and magnetic fields exhibit significant variations. Still, these variable energy fluxes satisfy several exact relations due to conservation of energy. In this paper, using numerical simulations, we quantify the variable energy fluxes of MHD turbulence, as well as verify several exact relations. We also study the energy fluxes of Elsässer variables that are constant in the inertial range.

**Keywords:** MHD turbulence; energy flux; variable energy flux; direct numerical simulation; exact relations; energy spectrum





## 1. Introduction

Magnetohydrodynamics (MHD) provides a framework to study the dynamics of flows in interstellar medium, galaxies, accretion disks, stars and planet interiors, solar wind, Tokamak, etc. [1]. In these systems, typically, the kinetic and magnetic Reynolds numbers are significantly large; hence, they exhibit turbulent behavior. The three-dimensional (3D) MHD turbulence and the 3D hydrodynamics turbulence have several common features. For example, both these systems exhibit nonlinear interactions among multiple scales. In addition, the energy injected at large scales cascades to intermediate and then to small scales. The dissipative terms destroy the fluid energy at small scales [2]. This cascade is quantified by *energy flux*.

Since inertial range of hydrodynamic turbulence does not have any forcing (except very weak viscous force), the energy flux in the inertial range remains constant [2–4]. However, MHD turbulence has a magnetic field in addition to the velocity field, and there are energy exchanges among the two fields via nonlinear interactions. Consequently, there are many energy fluxes, e.g., velocity to velocity, magnetic to magnetic and velocity to magnetic. Dar et al. [5] and Verma [6] showed that MHD turbulence has six energy fluxes related to the velocity and magnetic fields. Due to the energy exchange between the velocity and magnetic fields, the energy flux for the velocity field is no longer constant. Verma et al. [7] showed that in a typical turbulent magnetofluid, the inertial-range kinetic energy flux is depleted due to the energy transfer from the velocity to the magnetic field. The magnetic energy flux also exhibits variability due to these transfers.

MHD turbulence is also described in terms of Elsässer variables, which are $\mathbf{z}^{\pm} = \mathbf{u} \pm \mathbf{b}$. The fluxes associated with the Elsässer variables are constant in the inertial range [8,9]. This is because there is no energy exchange between the $\mathbf{z}^{+}$ and $\mathbf{z}^{-}$ variables. The mean magnetic field also affects the energy fluxes. In this paper, we study the energy fluxes in the presence of nonzero cross helicity, but no magnetic helicity and mean magnetic field.

There are several models for MHD turbulence that describe the associated energy spectra and fluxes. Kraichnan [10], Iroshnikov [11] and Dobrowolny et al. [12] assumed

the turbulence to be homogeneous and isotropic and argued that the kinetic and magnetic energy spectra scale as $E(k) \approx (\epsilon B_0)^{1/3} k^{-3/2}$, where $B_0$ is the magnitude of the mean magnetic field or of the large-scale magnetic field. Marsch [13] argued that in the absence of a mean magnetic field, the energy spectra of Elsässer variables exhibit a $k^{-5/3}$ spectra. Using renormalization group theory, Verma [6,14] argued that the *effective mean magnetic field* scales as $k^{-1/3}$, and hence the energy spectra of kinetic and magnetic fields scale as $E(k) \approx k^{-5/3}$. Goldreich and Sridhar [15] argued that in the presence of a strong external magnetic field, the flow becomes anisotropic and that $E(k_\perp) \approx k_\perp^{-5/3}$, where $k_\perp$ is the wavenumber perpendicular to the mean magnetic field. In this paper, we do not discuss the energy spectrum of MHD turbulence in detail, but focus on the energy fluxes.

Some of the important past works related to energy fluxes of MHD turbulence are as follows. Dar et al. [5] formulated the energy fluxes for MHD turbulence in terms of *mode-to-mode energy transfers* and computed the fluxes for two-dimensional (2D) MHD turbulence. Debliquy et al. [16], Alexakis et al. [17] and Carati et al. [18] computed these fluxes, as well as *shell-to-shell energy transfers*, for 3D MHD turbulence. Note that the simulations of Debliquy et al. [16] are for decaying turbulence. Using numerical simulations, Verma et al. [19] computed the energy fluxes of $z^\pm$ and showed consistency with the predictions of Marsch [13]. In the presence of mean magnetic field, Teaca et al. [20] and Sundar et al. [21] computed the corresponding energy fluxes. Verma [6,22,23,24] computed the energy fluxes using field-theoretic formalism. In addition, Verma [9] also described several exact relations among these fluxes using the analytical formalism of variable energy flux.

Solar wind provides an important platform to test the theories of MHD turbulence. Matthaeus and Goldstein [25] and Tu and Marsch [26] analyzed the solar wind data and observed a near $k^{-5/3}$ energy spectra for $u, b, z^\pm$ fields. Parashar et al. [27] studied the variations of spectral indices as a function of cross helicity. Verma et al. [28] estimated the energy flux in the solar wind using the energy spectrum and assuming Kolmogorov-like turbulence phenomenology for the MHD turbulence [13]. A more commonly-used strategy for the estimation of energy flux in the solar wind is to employ structure functions and the four-third rule [29]; Sorriso-Valvo et al. [30] employed this method. Following a similar procedure, Bandyopadhyay et al. [31] computed the energy flux near the Sun and argued that energy flux is enhanced here.

The effects of the magnetic field on the dynamics of peristaltic and nanofluid flows have been extensively studied in literature [32–37]. Eldesoky et al. [34] investigated the combined effects of the magnetic field and heat transport in peristaltic flow, and showed that the magnetic field enhances the thermal energy of the fluid. It has been argued that the magnetic field also affects the dynamics of biological nanofluids, which could have practical implications. Abdlesalam and Sohail [36] showed that the velocity distribution of the bioconvective flow of viscous fluid reduces with an enhancement of the magnetic field. An analytical investigation of Abdelsalam, Velasco-Hernández and Zaher [37] showed that the propulsive velocity of swimming sperms increases with the Hartmann number, a measure of the strength of the magnetic field. Apart from the effect of the magnetic field, the dynamics of peristaltic flows and nanofluid flow are also affected by the particle concentration in the flow, viscosity of the flow and temperature [38–40].

In MHD turbulent flow, the flow properties are usually studied using the Newtonian model, where the relation between stress and strain rate is linear [17,18,41]. However, there are flows in nature, where the relation of stress and strain rate are nonlinear. Such flows are studied using non-Newtonian models like Williamson flow, Casson Carreau and Jeffrey flow, etc. The dynamics of non-Newtonian flows and the flows through the porous medium like Darcy–Forchheimer flow are also affected by the magnetic field and have been broadly investigated in the literature [42–48]. A numerical investigation of MHD Williamson nanofluid by Rasool et al. showed that the drag force exerted by the medium on the flow increases with the magnetic parameter, a measure of the strength of magnetic field [43]. Rasool et al. [48] investigated the dynamics of convective MHD nanofluid flow

bounded by non-linear stretching surface and reported a decrease in flow velocity with an increase in magnetic parameter. Ali et al. [45] reported a decrease of Nusselt number with an increase in magnetic parameter for Darcy–Forchheimer rotating flow of a Casson Carreau nanofluid.

In this paper, we compute the various energy fluxes of forced MHD turbulence and validate several exact relations with numerical results. We perform a direct numerical simulation of forced MHD turbulence on a $512^3$ grid with kinetic hyperviscosity and magnetic hyperdiffusivity. We set random initial conditions for both the velocity and magnetic fields, and inject kinetic energy at wavenumber shell $(2, 3)$. We observe that the numerical results satisfy several exact relations of the fluxes in MHD turbulence. We also find that the fluxes of the Elsässer variables are constant in the inertial range.

The outline of this paper is as follows. In Section 2, we present various energy fluxes of MHD turbulence and the exact relations relating them. In Sections 3 and 4, we present the details of our numerical simulation and verification of exact results. Finally, we conclude in Section 5.

## 2. Energy Fluxes and Exact Relations

The equations for incompressible MHD turbulence in the absence of a mean magnetic field are [49]

$$\frac{\partial \mathbf{u}}{\partial t} + (\mathbf{u} \cdot \boldsymbol{\nabla})\mathbf{u} = -\boldsymbol{\nabla} p + \mathbf{F}_u(\mathbf{b}, \mathbf{b}) + \nu_0 \nabla^2 \mathbf{u} + \mathbf{F}_{ext}, \tag{1}$$

$$\frac{\partial \mathbf{b}}{\partial t} + (\mathbf{u} \cdot \boldsymbol{\nabla})\mathbf{b} = \mathbf{F}_b(\mathbf{b}, \mathbf{u}) + \eta_0 \nabla^2 \mathbf{b}, \tag{2}$$

$$\boldsymbol{\nabla} \cdot \mathbf{u} = 0, \tag{3}$$

$$\boldsymbol{\nabla} \cdot \mathbf{b} = 0, \tag{4}$$

where $\mathbf{u}$ and $\mathbf{b}$ are respectively the velocity and magnetic fields, $p$ is the total pressure (a sum of kinetic and magnetic pressures); $\nu_0$ and $\eta_0$ are respectively the viscosity and diffusivity. The random large-scale force term is $\mathbf{F}_{ext}$, and

$$\mathbf{F}_u = (\mathbf{b} \cdot \nabla)\mathbf{b}, \tag{5}$$

$$\mathbf{F}_b = (\mathbf{b} \cdot \nabla)\mathbf{u} \tag{6}$$

represent respectively the Lorentz force and the stretching of the magnetic field by the velocity field. Note that the magnetic field $\mathbf{b}$ is normalized using $\mathbf{b} = \mathbf{b}_{cgs} / \sqrt{4\pi\rho}$ to convert it in the units of velocity; here, $\mathbf{b}_{cgs}$ is the magnetic field in cgs units, and $\rho$ is the fluid density.

The scale-dependent properties of a system are customarily studied using Fourier transforms. In Fourier space, the evolution equations for the kinetic and magnetic modal energies are [8]

$$\frac{\partial E_u(\mathbf{k}, t)}{\partial t} = T_{uu}(\mathbf{k}, t) + \mathcal{F}_{ub}(\mathbf{k}, t) - D_u(\mathbf{k}, t) + \mathcal{F}_{ext}(\mathbf{k}, t), \tag{7}$$

$$\frac{\partial E_b(\mathbf{k}, t)}{\partial t} = T_{bb}(\mathbf{k}, t) + \mathcal{F}_{bu}(\mathbf{k}, t) - D_b(\mathbf{k}, t), \tag{8}$$

where $\mathbf{u}(\mathbf{k})$ and $\mathbf{b}(\mathbf{k})$ are respectively the Fourier transforms of the velocity and magnetic fields, $E_u(\mathbf{k}) = |\mathbf{u}(\mathbf{k})^2|/2$ and $E_b(\mathbf{k}) = |\mathbf{b}(\mathbf{k})^2|/2$ are respectively the modal kinetic and magnetic energies, $T_{uu}(\mathbf{k}, t)$ and $T_{bb}(\mathbf{k}, t)$ are nonlinear energy transfers arising due to $(\mathbf{u} \cdot \nabla)\mathbf{u}$ and $(\mathbf{u} \cdot \nabla)\mathbf{b}$, respectively, $D_u(\mathbf{k})$ and $D_b(\mathbf{k})$ are respectively the kinetic and magnetic energy dissipation rates, $\mathcal{F}_{ext}(\mathbf{k}, t)$ is the energy injection rate due the external force and $\mathcal{F}_{ub}(\mathbf{k})$ and $\mathcal{F}_{bu}(\mathbf{k})$ are the cross energy transfers, i.e., from magnetic to kinetic and vice versa. The formulas for the above transfers are as follows [8]:

$$\mathcal{F}_{ub}(\mathbf{k}) = -\sum_{\mathbf{p}} \Im\{[\mathbf{k}\cdot\mathbf{b}(\mathbf{q})][\mathbf{b}(\mathbf{p})\cdot\mathbf{u}^*(\mathbf{k})]\}, \tag{9}$$

$$\mathcal{F}_{bu}(\mathbf{k}) = -\sum_{\mathbf{p}} \Im\{[\mathbf{k}\cdot\mathbf{b}(\mathbf{q})][\mathbf{u}(\mathbf{p})\cdot\mathbf{b}^*(\mathbf{k})]\}, \tag{10}$$

$$T_{uu}(\mathbf{k}) = \sum_{\mathbf{p}} \Im\{[\mathbf{k}\cdot\mathbf{u}(\mathbf{q})][\mathbf{u}(\mathbf{p})\cdot\mathbf{u}^*(\mathbf{k})]\}, \tag{11}$$

$$T_{bb}(\mathbf{k}) = \sum_{\mathbf{p}} \Im\{[\mathbf{k}\cdot\mathbf{u}(\mathbf{q})][\mathbf{b}(\mathbf{p})\cdot\mathbf{b}^*(\mathbf{k})]\}, \tag{12}$$

$$\mathcal{F}_{ext} = \Re[\mathbf{F}_{ext}(\mathbf{k})\cdot\mathbf{u}^*(\mathbf{k})]. \tag{13}$$

where $\Im$, $\Re$ and $^*$ represent the imaginary and real parts, and the conjugate of a complex number, respectively. In addition, $\mathbf{q} = \mathbf{k} - \mathbf{p}$.

The kinetic and magnetic energy spectra are computed using

$$E_u(k) = \frac{1}{2}\sum_{k\leq|\mathbf{k}'|<k+1} |\mathbf{u}(\mathbf{k}')|^2, \tag{14}$$

$$E_b(k) = \frac{1}{2}\sum_{k\leq|\mathbf{k}'|<k+1} |\mathbf{b}(\mathbf{k}')|^2. \tag{15}$$

Apart from the kinetic and magnetic energy spectra, we also have energy spectra related to the Elsässer variables:

$$E_{z^\pm} = \frac{1}{2}\sum_{k\leq|\mathbf{k}'|<k+1} |\mathbf{z}^\pm(\mathbf{k}')|^2, \tag{16}$$

where $\mathbf{z}^\pm(\mathbf{k}) = \mathbf{u}(\mathbf{k}) \pm \mathbf{b}(\mathbf{k})$ are the Elsässer fields.

In MHD turbulence, there are four nonlinear terms in the governing equations. These terms yield six energy fluxes which are illustrated in Figure 1b and are defined as follows [5,6,8]:

$$\Pi_{u_>}^{u_<}(k_0) = \sum_{k>k_0}\sum_{p\leq k_0} \Im\{[\mathbf{k}\cdot\mathbf{u}(\mathbf{q})][\mathbf{u}(\mathbf{p})\cdot\mathbf{u}^*(\mathbf{k})]\}, \tag{17}$$

$$\Pi_{b_>}^{u_<}(k_0) = -\sum_{k>k_0}\sum_{p\leq k_0} \Im\{[\mathbf{k}\cdot\mathbf{b}(\mathbf{q})][\mathbf{u}(\mathbf{p})\cdot\mathbf{b}^*(\mathbf{k})]\}, \tag{18}$$

$$\Pi_{b_<}^{u_>}(k_0) = -\sum_{k\leq k_0}\sum_{p>k_0} \Im\{[\mathbf{k}\cdot\mathbf{b}(\mathbf{q})][\mathbf{u}(\mathbf{p})\cdot\mathbf{b}^*(\mathbf{k})]\}, \tag{19}$$

$$\Pi_{b_>}^{b_<}(k_0) = \sum_{k>k_0}\sum_{p\leq k_0} \Im\{[\mathbf{k}\cdot\mathbf{u}(\mathbf{q})][\mathbf{b}(\mathbf{p})\cdot\mathbf{b}^*(\mathbf{k})]\}, \tag{20}$$

$$\Pi_{b_<}^{u_<}(k_0) = \sum_{k\leq k_0}\sum_{p\leq k_0} \Im\{[\mathbf{k}\cdot\mathbf{b}(\mathbf{q})][\mathbf{u}(\mathbf{p})\cdot\mathbf{b}^*(\mathbf{k})]\}, \tag{21}$$

$$\Pi_{b_>}^{u_>}(k_0) = \sum_{k>k_0}\sum_{p>k_0} \Im\{[\mathbf{k}\cdot\mathbf{b}(\mathbf{q})][\mathbf{u}(\mathbf{p})\cdot\mathbf{b}^*(\mathbf{k})]\}, \tag{22}$$

where the superscript and subscript of $\Pi$ represent respectively the giver and receiver modes, and $<$ and $>$ denote respectively the modes inside and outside the sphere of radius $k_0$. For example, $\Pi_{Y_>}^{X_<}(k_0)$ denotes the rate of energy transfer from the wavenumbers inside the sphere of radius $k_0$ of field $X$ to the wavenumbers outside the sphere of field $Y$. It is easy to show that $\Pi_Y^X(k_0) = -\Pi_X^Y(k_0)$ [5].

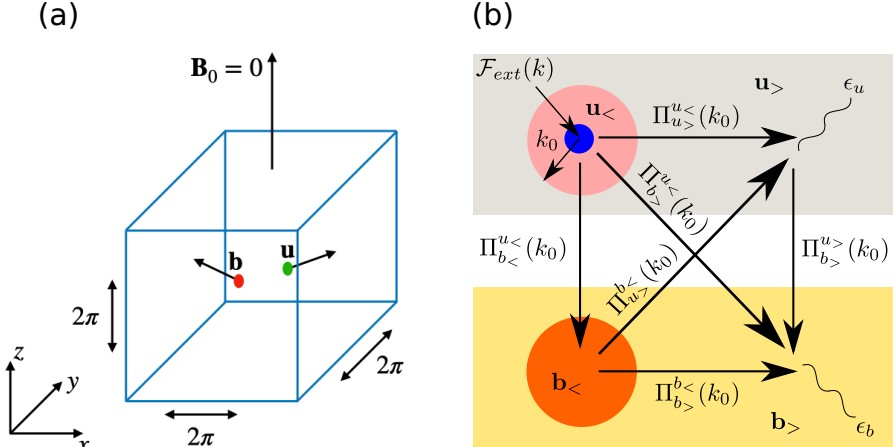

**Figure 1.** Schematic diagram of (**a**) a periodic simulation box where **u** and **b** respectively represent random fluctuations in velocity and magnetic field, and (**b**) various energy fluxes in MHD turbulence. These energy fluxes are destroyed in the dissipation range with viscous dissipation rate $\epsilon_u$ and magnetic dissipation rate $\epsilon_b$.

Note that there are energy exchanges among the velocity field and the magnetic field. However, there is no such cross transfer between $\mathbf{z}^+$ and $\mathbf{z}^-$. For these variables, the corresponding energy fluxes are

$$\Pi_{z_>^+}^{z_<^+}(k_0) = \sum_{k>k_0} \sum_{p \leq k_0} \Im\{[\mathbf{k} \cdot \mathbf{z}^-(\mathbf{q})][\mathbf{z}^+(\mathbf{p}) \cdot \mathbf{z}^{+*}(\mathbf{k})]\}, \tag{23}$$

$$\Pi_{z_>^-}^{z_<^-}(k_0) = \sum_{k>k_0} \sum_{p \leq k_0} \Im\{[\mathbf{k} \cdot \mathbf{z}^+(\mathbf{q})][\mathbf{z}^-(\mathbf{p}) \cdot \mathbf{z}^{-*}(\mathbf{k})]\}. \tag{24}$$

The total energy flux in MHD turbulence, which is a sum of energy transfers from the velocity and magnetic modes inside the sphere to the modes outside the sphere, is

$$\Pi_{total}(k_0) = \Pi_{u_>}^{u_<}(k_0) + \Pi_{b_>}^{u_<}(k_0) + \Pi_{u_>}^{b_<}(k_0) + \Pi_{b_>}^{b_<}(k_0). \tag{25}$$

It can be easily shown that $2\Pi_{total}(k_0) = \Pi_{z_>^+}^{z_<^+}(k_0) + \Pi_{z_>^-}^{z_<^-}(k_0)$.

In 3D hydrodynamic turbulence, the kinetic energy flux $\Pi_{u_>}^{u_<}$ remains constant in inertial range and it is equal to the kinetic energy dissipation rate. This flux, however, is not constant in MHD turbulence [7]. However, the total energy transferred to the inertial range of the velocity field is dissipated by the viscous force. This leads to

$$\Pi_{u_>}^{u_<} + \Pi_{u_>}^{b_<} + \Pi_{u_>}^{b_>} = \epsilon_u, \tag{26}$$

where $\epsilon_u$ is the kinetic energy dissipation rate. Refer to Figure 1b for an illustration. In the above expression, $\Pi_{u_>}^{b_<} + \Pi_{u_>}^{b_>}$ is the total energy transferred from the magnetic field to the velocity field of the inertial range. Note that each of the above fluxes, $\Pi_{u_>}^{u_<}$, $\Pi_{u_>}^{b_<}$ and $\Pi_{u_>}^{b_>}$, vary with $k$, but the sum is an approximate constant, hence, we call them *variable energy fluxes* [8,9].

In a similar vein, we derive the following relation for the energy flowing into the magnetic-field channel:

$$\Pi_{b_>}^{b_<} + \Pi_{b_>}^{u_<} + \Pi_{b_>}^{u_>} = \epsilon_b, \tag{27}$$

where $\epsilon_b$ is the magnetic energy dissipation rate. In addition, the energy injected into the large-scale velocity field is transferred to the inertial-range velocity and magnetic fields. These fluxes are dissipated at small scales. Consequently,

$$\Pi_{u_>}^{u_<} + \Pi_{u_<}^{b_<} + \Pi_{b_>}^{u_<} = \epsilon_u + \epsilon_b = \epsilon, \tag{28}$$

where $\epsilon$ is the total dissipation rate, which is equal to the total-energy injection rate. Similar balance of energy transfers in the magnetic channel leads to

$$\Pi_{b_<}^{u_<} + \Pi_{b_<}^{u_>} = \Pi_{b_>}^{b_<}. \tag{29}$$

Schematic diagram of Figure 1b helps us understand the above relations.

The above four equations represent *exact relations*. The equality holds statistically in spite of the fact that each of the fluxes exhibits significant variations with $k$. In the next section, we validate the above relations using numerical simulations.

## 3. Governing Equations and Simulation Method

We employ a pseudo-spectral code named TARANG [50,51] to solve the following equations in a 3D cubic periodic box of size $(2\pi)^3$ (as shown in Figure 1a):

$$\frac{\partial \mathbf{u}}{\partial t} + (\mathbf{u} \cdot \boldsymbol{\nabla})\mathbf{u} = -\boldsymbol{\nabla} p + \mathbf{F}_u(\mathbf{b}, \mathbf{b}) + \nu \nabla^4 \mathbf{u} + \mathbf{F}_{ext}, \tag{30}$$

$$\frac{\partial \mathbf{b}}{\partial t} + (\mathbf{u} \cdot \boldsymbol{\nabla})\mathbf{b} = \mathbf{F}_b(\mathbf{b}, \mathbf{u}) + \eta \nabla^4 \mathbf{b}, \tag{31}$$

$$\boldsymbol{\nabla} \cdot \mathbf{u} = 0, \tag{32}$$

$$\boldsymbol{\nabla} \cdot \mathbf{b} = 0. \tag{33}$$

In the above equations, we employ the hyperviscous and hyperdiffusive terms to increase the inertial range and to suppress the dissipation range. In addition, the velocity, length and time are non-dimensionalized using characteristic velocity ($U_0$), box size ($2\pi$), and the eddy turn over time ($2\pi/U_0$). We use a fourth-order Runge–Kutta scheme for time marching and the Courant–Friedrich–Lewis (CFL) condition for optimizing the time step $\Delta t$. We employ 2/3 rule for idealizing.

We performed a numerical simulation on a $512^3$ grid with the hyperviscous and hyperdiffusion parameters $\nu = \eta = 3 \times 10^{-7}$. We use the Craya–Herring basis [52,53] to generate the random initial conditions for both velocity and magnetic field. In a Craya–Herring basis, the three basis vectors are,

$$\hat{e}_3(\mathbf{k}) = \hat{k}; \quad \hat{e}_1(\mathbf{k}) = (\hat{k} \times \hat{n})/|\hat{k} \times \hat{n}|; \quad \hat{e}_2(\mathbf{k}) = \hat{k} \times \hat{e}_1(\mathbf{k}), \tag{34}$$

where $\hat{n}$ is chosen as any arbitrary direction, and $\hat{k}$ is the unit vector for wavenumber $\mathbf{k}$. The velocity field for 3D incompressbile flow in the Craya–Herring basis is written as

$$\mathbf{u}(\mathbf{k}) = u_1(\mathbf{k})\hat{e}_1(\mathbf{k}) + u_2(\mathbf{k})\hat{e}_2(\mathbf{k}). \tag{35}$$

We start our simulation with the following initial condition:

$$u_1(\mathbf{k}) = \sqrt{(E_u/2N^3)} \; i(\exp(i\phi_1(\mathbf{k})) - \exp(i\phi_2(\mathbf{k}))), \tag{36}$$

$$u_2(\mathbf{k}) = \sqrt{(E_u/2N^3)} \; (\exp(i\phi_1(\mathbf{k})) + \exp(i\phi_2(\mathbf{k}))), \tag{37}$$

where $N^3$ is the total number of modes, $E_u = 0.5$ is the total kinetic energy and the phases $\phi_1(\mathbf{k})$ and $\phi_2(\mathbf{k})$ are chosen randomly in the band $[0, 2\pi]$. Then we transform the velocity field from Craya–Herring basis to Cartesian basis. We use a similar approach to generate the random initial condition for magnetic field with total initial magnetic energy set as 0.25.

Note that the mean magnetic field in our simulation is zero, i.e., $\mathbf{B}_0 = 0$. We define Reynolds number Re = $u_{rms}L^3/\nu$, and magnetic Reynolds number Rm = $u_{rms}L^3/\eta$, where $L = 2\pi$ is the box size [54]. The $L^3$ factor arises due to the $\nabla^4$ factor of the hyperdiffusion term. For the steady state, both these numbers are $8.2 \times 10^8$. However, we caution that they are not reliable estimates of Reynolds number and magnetic Reynolds number, and they should not to be compared with earlier simulations.

We apply random force to all the velocity modes in a wavenumber shell $(2, 3)$, which is denoted by $k_f = 2$. We follow the procedure outlined in Alvelius [55] and Maffioli [56]. Note that we force only the velocity modes. The kinetic energy injection rate is 0.4. The simulation parameters are tabulated in Table 1. In addition, the simulation workload is summarized in Table S1 and Figure S1 of the Supplementary Material.

**Table 1.** Simulation parameters: Grid resolution ($N$), energy injection rate ($\epsilon$), kinematic hyperviscosity ($\nu$), magnetic hyperdiffusivity ($\eta$), Magnetic Prandtl number (Pm), kinetic Reynolds number Re = $u_{rms}L^3/\nu$ and magnetic Reynolds number Rm = $u_{rms}L^3/\eta$, where $L = 2\pi$ is the box size.

| $N$ | $\epsilon$ | $\nu$ | $\eta$ | Pm | Re | Rm |
|---|---|---|---|---|---|---|
| $512^3$ | 0.4 | $3 \times 10^{-7}$ | $3 \times 10^{-7}$ | 1 | $8.2 \times 10^8$ | $8.2 \times 10^8$ |

In Figure 2a, we exhibit the time series of the total kinetic energy (solid red curve), the total magnetic energy (solid green curve) and the total energy (solid blue curve). Figure 2b exhibits the corresponding dissipation rates. Clearly, the system reaches a steady state in 2 to 3 eddy turnover time. During the steady state, the magnetic energy dissipation rate is larger than the kinetic energy dissipation rate and the total dissipation rate matches with the energy injection rate (Figure 2b). Interestingly, the kinetic energy and the magnetic energy are equipartitioned during the steady state.

In the next section we will describe the numerical results related to the energy fluxes.

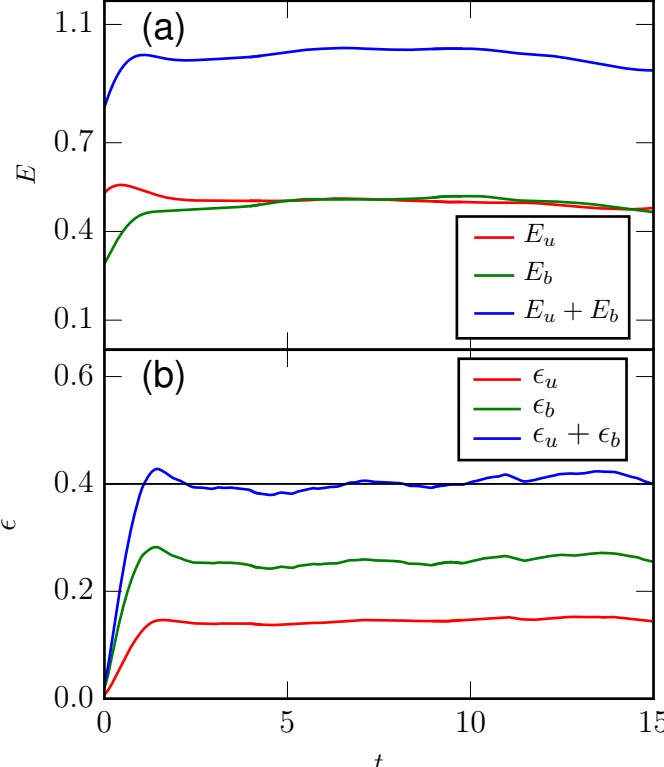

**Figure 2.** The evolution of (**a**) kinetic energy (solid red curve), magnetic energy (solid green curve) and total energy (solid blue curve), and (**b**) kinetic, magnetic and total dissipation rates with the same color scheme. Note that the solid black curve shows the total-energy injection rate $\epsilon_{\text{inj}} = 0.4$.

## 4. Numerical Results on the Energy Fluxes

Before embarking on energy flux studies, we compute the spectra of the velocity and magnetic fields. We observe that the spectra show significant fluctuations; hence, to obtain smooth curves in the inertial range, we compute the cumulative spectra, which is $\sum_k^\infty E_X(k')$, where $X = u, b, z^\pm$. Note that if the inertial-range energy spectrum scales as $k^\alpha$, then the cumulative spectrum would scale as $k^{\alpha+1}$. In Figure 3, we exhibit the cumulative spectra of the velocity, magnetic and $z^\pm$ variables averaged over time interval 9 to 12 (in non-dimensional time units).

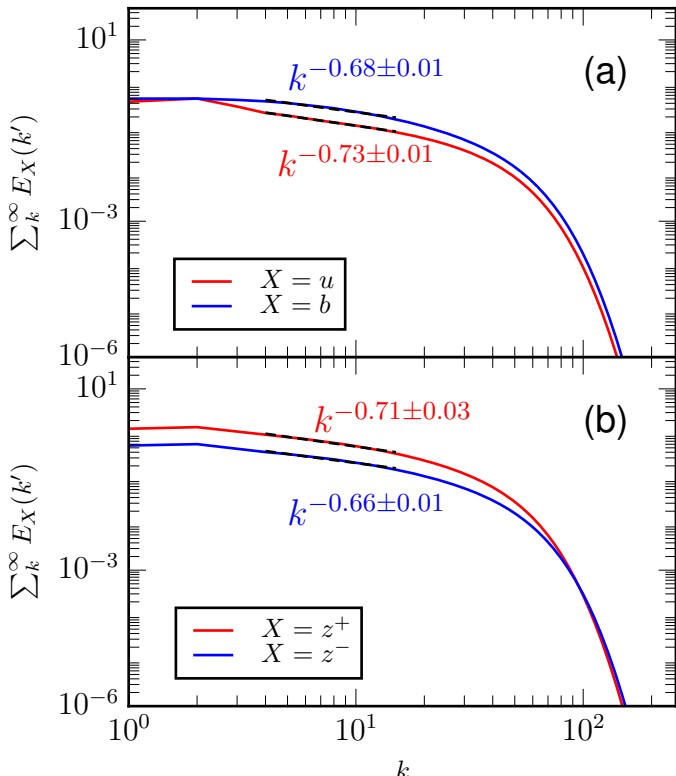

**Figure 3.** Time-averaged cumulative spectra of (**a**) the kinetic energy (solid red curve) and magnetic energy (solid blue curve), and (**b**) $z^+$ (solid red curve) and $z^-$ (solid blue curve). These spectra are close to $k^{-2/3}$ indicating consistency with the Kolmogorov-like spectra for MHD turbulence.

As shown in Figure 3, in the wavenumber range of (3,20), the cumulative spectra follow power laws to a reasonable accuracy. The exponents of the power law for $u, b, z^+, z^-$ variables are $-0.73 \pm 0.01$, $-0.68 \pm 0.01$, $-0.71 \pm 0.03$ and $-0.66 \pm 0.01$ respectively, which are close to $-2/3$. Hence, we conclude that energy spectra for these fields are reasonably close to Kolmogorov's $k^{-5/3}$ power law. The deviations of the exponents from $-5/3$ may be related to the variable energy flux and intermittency [2,8]. Note that the kinetic energy spectrum, $\sim k^{-0.73}$, is steeper than the magnetic energy spectrum, $\sim k^{-0.68}$. This differential can be attributed to the energy transfer from the velocity field to the magnetic field [7]. We have compared our results with those from the past, e.g., observational works on solar wind [25,26] and numerical works [57–60]. We observed general consistency between these results. Note, however, that several authors report $k^{-3/2}$ energy spectra, but a detailed discussion on this topic will take us beyond the scope of the paper. We also remark that higher-resolution simulations, which are planned in future, would provide a better handle on the spectral exponents.

Following the main objectives of the paper, we compute various energy fluxes of MHD turbulence for 80 concentric spheres. The radii of the first 16 spheres are linearly

binned as [1, 2, 3,..., 16], and those of the last two spheres are 128 and 256. The radii of the intermediate spheres are binned as follows:

$$r_i = r_{16} 2^{s(i-16)},\tag{38}$$

where $r_{16} = 16$, $s = \log_2(r_{max}/32)/(n-5)$, with $r_{max}$ being the radius of the largest wavenumber sphere, and $n$ as the total number of spheres.

In Figure 4, we exhibit the plots of various energy fluxes, which are averaged over a time interval of $t = 9$ to 12. The black solid curve represents the total-energy injection rate. These plots reveal the following interesting properties of the energy fluxes of MHD turbulence:

1.  The energy fluxes corresponding to the total energy and $z^{\pm}$ are nearly constant in the wavenumber band (3, 20), consistent with the power-law regime of the energy spectra discussed earlier. Note that the inertial-range energy flux of the total energy matches with $[\Pi_{z^+} + \Pi_{z^-}]/2$; in addition, these fluxes are equal to the energy supply rate and the total-energy dissipation rate, consistent with the conservation of energy.

2.  As shown in Figure 4a, in the wavenumber band $k = (3, 6)$, the kinetic energy flux dips sharply, while the magnetic energy fluxes, $\Pi_{b_>}^{u_<}$ and $\Pi_{b_<}^{u_<}$, grow rapidly. This observation indicates energy transfer from $u$ to $b$. Note that $\Pi_{b_>}^{b_<}$ picks up significantly after this band $(k = (3, 6))$.

3.  The energy fluxes $\Pi_{b_<}^{u_>}$ and $\Pi_{b_>}^{u_>}$ are negative and become significant beyond wavenumber band (3, 6). These fluxes indicate energy transfers from the magnetic field to the intermediate-scale velocity field. Consequently, $\Pi_{u_>}^{u_<}$ grows and becomes significant beyond $k = 10$.

4.  The energy fluxes corresponding to the velocity and magnetic fields exhibit significant variability due to cross energy transfers. However, the fluxes of $z^{\pm}$ are nearly constant in the inertial range due to lack of such transfers. We also compute the flux of cross helicity, which is $\Pi_{Hc}(k) = (\Pi_{z^+}(k) - \Pi_{z^-}(k))/4$, and exhibit this flux in Figure 4b. We need to further explore the evolution of cross helicity flux in MHD turbulence.

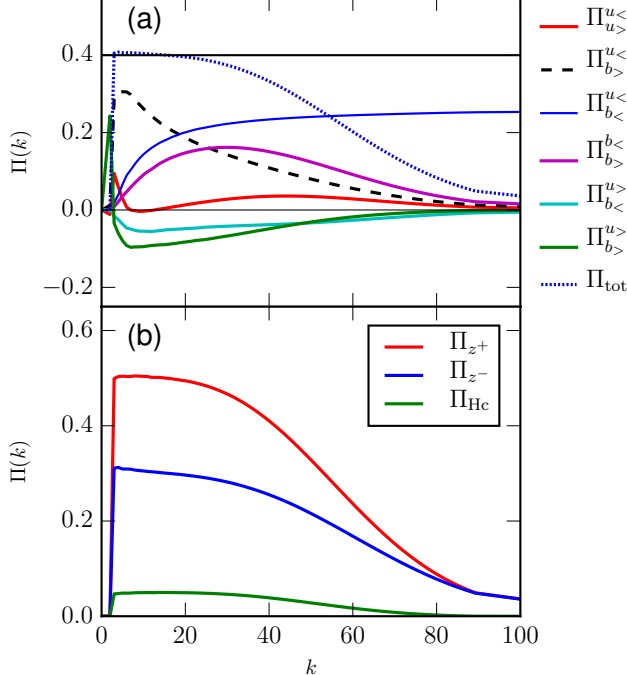

**Figure 4.** Plots of various fluxes in MHD turbulence (**a**) $\Pi_{u_>}^{u_<}$, $\Pi_{b_>}^{u_<}$, $\Pi_{b_<}^{u_>}$, $\Pi_{b_<}^{u_<}$, $\Pi_{b_>}^{b_<}$, $\Pi_{b_>}^{u_>}$ and $\Pi_{total}$, and (**b**) $\Pi_{z^+}$, $\Pi_{z^-}$ and $\Pi_{Hc}$ averaged over time frame $t = 9 - 12$. Note, the solid black curve shows the total-energy injection rate.

We compare the results of energy fluxes of MHD turbulence with the earlier works reported in literature [17,18,41]. Our results for the transfer of energy between magnetic field, and the transfer of energy from the velocity field to the magnetic field are consistent with those of Alexakis, Mininni and Pouquet [17], Carati et al. [18] and Alexakis, Mininni and Pouquet [41].

Finally, we come to the exact relations for the energy fluxes of MHD turbulence. The four subfigures of Figure 5 provide numerical verification of Equations (26)–(29) as follows:

1.  Figure 5a demonstrates that in the inertial range, the sum $\Pi^{u<}_{u>} + \Pi^{b<}_{u>} + \Pi^{b>}_{u>}$ matches with the kinetic energy dissipation rate $\epsilon_u$. Note that the sum represents the total energy transfer to the inertial-range velocity modes that gets dissipated in the dissipation range; this is the reason for the equality of Equation (26).
2.  Figure 5b illustrates a similar balance between the energy transfer to the inertial-range magnetic modes and the magnetic-energy dissipation rate $\epsilon_b$ (see Equation (27)).
3.  The energy supplied to the large-scale velocity modes get transferred to the inertial-range velocity and magnetic modes. It leads to the exact relation of Equation (28). This relation is verified in Figure 5c.
4.  The magnetic field is not forced externally. Instead, the large-scale magnetic modes ($b_<$) receive energy from the velocity modes as $\Pi^{u<}_{b<} + \Pi^{u>}_{b<}$. The energy received by the large-scale magnetic modes cascades to the inertial range of the magnetic field as $\Pi^{b<}_{b>}$. Hence, $\Pi^{u<}_{b<} + \Pi^{u>}_{b<} = \Pi^{b<}_{b>}$. This relation is verified for the wavenumber range $k = (3, 18)$. See Figure 5d for an illustration.

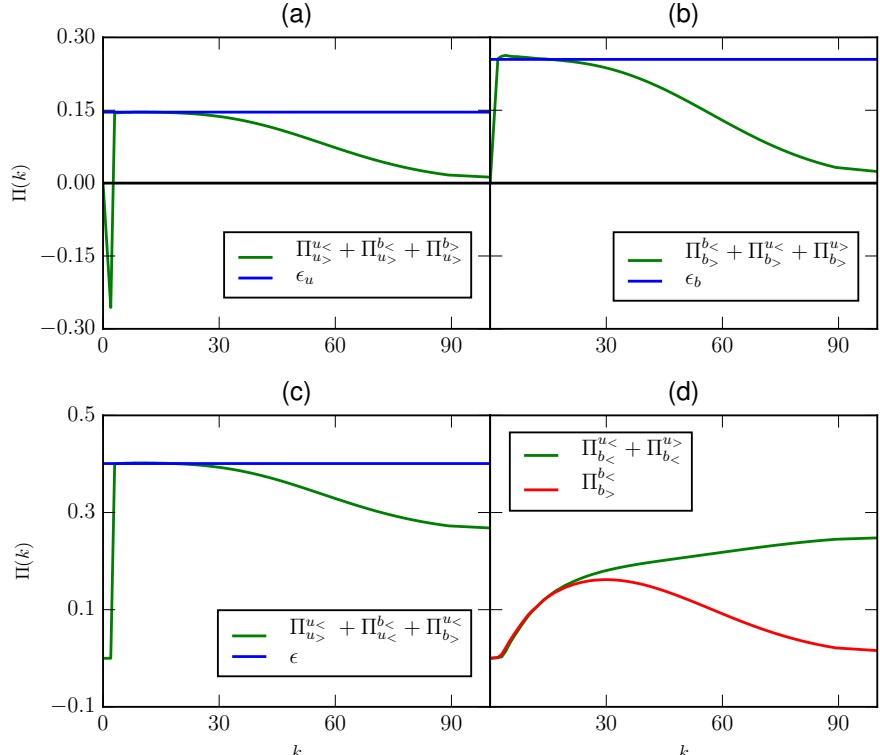

**Figure 5.** Plots of exact relations observed in the inertial range: (**a**) $\Pi^{u<}_{u>} + \Pi^{b<}_{u>} + \Pi^{b>}_{u>}$ (solid green curve) and $\epsilon_u$ (solid blue curve), (**b**) $\Pi^{b<}_{b>} + \Pi^{u<}_{b>} + \Pi^{u>}_{b>}$ (solid green curve) and $\epsilon_b$, (**c**) $\Pi^{u<}_{u>} + \Pi^{b<}_{u<} + \Pi^{u<}_{b>}$ (solid green curve) and $\epsilon = \epsilon_u + \epsilon_b$ (solid blue curve) and (**d**) $\Pi^{u<}_{b<} + \Pi^{u>}_{b<}$ (solid green curve) and $\Pi^{b<}_{b>}$ (solid red curve). The results are averaged over time frame $t = 9$–12.

Thus, we verify several important exact relations regarding the energy fluxes of MHD turbulence.

## 5. Conclusions

In MHD turbulence, there are complex energy transfers among the velocity and magnetic fields. The energy fluxes of MHD turbulence provide a measure for these transfers. In the past, these fluxes have been computed using numerical simulations [5,16–18]. In this paper, we describe the subtle variations of these fluxes in the framework of variable energy flux. The energy fluxes related to the velocity and magnetic fields vary with wavenumber. However, several combinations of these fluxes are constant; these are the exact relations related to the energy fluxes.

In this paper, we describe variable energy fluxes and exact relations of MHD turbulence using numerical simulations. We summarize our results as follows:

1.  Our work is focused on the energy fluxes of forced MHD turbulence, in contrast to those of decaying MHD turbulence, studied earlier by Debliquy et al. [16]. A close comparison between the two sets of energy fluxes shows that the decaying and the forced MHD turbulence have several critical differences. For example, we observe positive $\Pi^{u<}_{b_<}$, while Debliquy et al. [16] reported negative $\Pi^{u<}_{b_<}$.

2.  We employed hyperviscous and hyperdiffusive terms in our simulation to increase the extent of the inertial range. For our simulation, the flux for the total energy is nearly constant in the inertial range, which is $k = (3, 20)$. The extent of our inertial range is larger than that of Debliquy et al. [16], who do not employ hyperdiffusion.

3.  For our numerical simulations, the spectral indices for $u, b$ and $z^{\pm}$ are close to $-5/3$, rather than $-3/2$ [10]. A word of caution, however, is that the inertial range is quite narrow due to the moderate resolution ($512^3$) of our simulation. For a better understanding of the spectral indices and the energy fluxes, we need a broader inertial range that is possible with high-resolution simulations; we plan for such simulations in the near future.

4.  It has been recently reported that the energy fluxes of MHD turbulence satisfy several exact relations [8,9]. These relations are based on energy conservation principles. In this paper, we validate four such exact relations, which are Equations (26)–(29).

In summary, we employ numerical simulations to understand the variations of inertial-range energy fluxes of MHD turbulence. We also demonstrate several exact relations related to energy fluxes. This study provides valuable insights into the dynamics of MHD turbulence.

**Supplementary Materials:** The following are available online at https://www.mdpi.com/article/10.3390/fluids6060225/s1. Table S1: Workload of the simulation: the grid-resolution $N$, the total number of processor $p$, data division along $x$ direction $p_{\mathrm{col}}$, data division along $y$ direction $p_{\mathrm{row}}$, total RAM used in GB $M_{\mathrm{RAM}}$, simulation time $t$ (eddy turnover time) and simulation ran time in hours $T$. Figure S1: Schematic diagram for the pencil decomposition of a 3D array. From Chatterjee et al. [51]. Reprinted with permission from Elsevier.

**Author Contributions:** All authors contributed equally to this work. In particular, M.V. conceptualized the formalism and wrote the first version of the computer program. S.C., M.S. and S.A. implemented the new forcing function in the code, and performed numerical simulations. The results were analyzed and the paper was written by all the authors. All authors have read and agreed to the published version of the manuscript.

**Funding:** This project was supported by the Indo-French project 6104-1 from CEFIPRA. S. Chatterjee is supported by INSPIRE fellowship (No. IF180094) of the Department of Science and Technology, India.

**Institutional Review Board Statement:** This paper has not been reviewed by any institutional board.

**Informed Consent Statement:** The paper does not employ any clinical research or personal data, hence no consent is required.

**Data Availability Statement:** The numerical data will be made available up on request.

**Acknowledgments:** The authors thank Franck Plunian and Rodion Stepanov for valuable discussions. Parts of the simulations were run on IIT Kanpur's HPC clusters and Shaheen II of KAUST supercomputing laboratory, Saudi Arabia (Project No. k1416).

**Conflicts of Interest:** The authors have no conflict of interest.

**Abbreviations**

The following abbreviations are used in this manuscript:

| MHD | Magnetohydrodynamics |
|-----|----------------------|
| 3D | Three dimensions |
| 2D | Two dimensions |

**Nomenclature**

| | |
|---|---|
| $\mathbf{k}, \mathbf{p}, \mathbf{q}$ | Fourier wavenumbers |
| $\mathbf{u}$ | Velocity field |
| $\mathbf{b}$ | Magnetic field |
| $\mathbf{z}^{\pm}$ | Elsässer variables |
| $t$ | Time |
| $p$ | Pressure |
| $\mathbf{F}_{ext}$ | Random large-scale force |
| $\mathbf{F}_u$ | Lorentz force |
| $\mathbf{F}_b$ | Stretching of magnetic field by velocity field |
| $E_u(\mathbf{k})$ | Modal kinetic energy |
| $E_b(\mathbf{k})$ | Modal magnetic energy |
| $E_u(k)$ | Kinetic energy spectrum |
| $E_b(k)$ | Magnetic energy spectrum |
| $E_{z^{\pm}}(k)$ | Elsässer energy spectra |
| $T_{uu}(\mathbf{k}), T_{bb}(\mathbf{k})$ | Nonlinear modal energy transfers |
| $D_u(\mathbf{k})$ | Modal kinetic energy dissipation rate |
| $D_b(\mathbf{k})$ | Modal magnetic energy dissipation rate |
| $\mathcal{F}_{ext}(\mathbf{k})$ | External modal energy injection rate |
| $\mathcal{F}_{ub}(\mathbf{k}), \mathcal{F}_{bu}(\mathbf{k})$ | Cross energy transfers among velocity and magnetic modes |
| $\epsilon_u$ | Kinetic energy dissipation rate |
| $\epsilon_b$ | Magnetic energy dissipation rate |
| $\epsilon$ | Total dissipation rate |
| $\epsilon_{\text{inj}}$ | Kinetic energy injection rate |
| $\Pi_{Y_>}^{X_<}$ | Energy flux from the wavenumbers inside the sphere of radius $k_0$ of field $X$ to outside the sphere of field of field $Y$, e.g., $X = Y = u, b, z^{\pm}$ |
| $(\hat{e}_1, \hat{e}_2, \hat{e}_3)$ | Unit vectors in Craya–Herring basis |
| $U_0$ | Characteristic velocity |
| $(u_1, u_2)$ | Velocity components in Craya–Herring basis |
| $\mathbf{B}_0$ | Mean magnetic field |
| $\Pi_{total}$ | Total energy flux |
| $L$ | Periodic box size used in simulation |
| $N$ | Grid size |
| $(\nu, \eta)$ | Kinematic hyperviscosity, magnetic hyperdiffusivity |
| $(\nu_0, \eta_0)$ | Kinematic viscosity, magnetic diffusivity |
| Re | Kinetic Reynolds number |
| Rm | Magnetic Reynolds number |
| Pm | Magnetic Prandtl number |
| $\sum_k^{\infty} E_X(k')$ | Cumulative spectra, where $X = u, b, z^{\pm}$ |
| $r_i$ | Radius of ith intermediate sphere |
| $n$ | Total no. of spheres |
| $r_{max}$ | Radius of target wavenumber sphere |

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
