# Peer review of "Variable Energy Fluxes and Exact Relations in Magnetohydrodynamics Turbulence"

_fluids, doi:10.3390/fluids6060225_

Round 1
Reviewer 1 Report
The article studies the variable energy fluxes of MHD turbulence, as well as verify several exact relations. They also study the energy fluxes of Elsässer variables that are constant in the inertial range.
The article has publishable material and my comments are as follows:
1- Similarity with other articles is high. Authors are invited to re-write their paper.
2- The first few lines in the abstract should belong to the introduction. Instead, please explain the method used, the aim for the proposed study, and the key results of the current research.
3- Compare the results with previous work for validation.
4- Reference list is not uniform. Author should use ISO abbreviation for Journals names. It should be as per required style of journal.
4- Improve the legend and quality of figures.
5- There are few language and typographic errors and some "glitches" of editing must be fixed in revised paper.
6- For fortifying the introduction section with the new publications, old references should be replaced with new such as:
Interaction between compressibility and particulate suspension on peristaltically driven flow in planar channel
A study of nonlinear variable viscosity in finite-length tube with peristalsis
Numerical approach of variable thermophysical features of dissipated viscous nanofluid comprising gyrotactic micro-organisms
Joint Effect of Magnetic Field and Heat Transfer on Particulate Fluid Suspension in a Catheterized Wavy Tube
Anomalous reactivity of thermo-bioconvective nanofluid towards oxytactic microorganisms
Hall and transverse magnetic field effects on peristaltic flow of a maxwell fluid through a porous medium
Adverse effects of a hybrid nanofluid in a wavy non-uniform annulus with convective boundary conditions
Electromagnetically modulated self-propulsion of swimming sperms via cervical canal
Thermal transport of radiative Williamson fluid over stretchable curved surface
Accordingly, I advise that the manuscript is put under major revision until the inquires addressed are responded to.
Reviewer 2 Report
This is a good piece of work and the manuscript is well-written.
Author Response
We thank the referee for the encouraging comments on our manuscript.
Reviewer 3 Report
Please see the attachment.
Round 2
Reviewer 1 Report
Authors have done the required amendments and article is ready for publication.
Reviewer 3 Report
Accept in present form.